# Evaluation of Antiviral Activity of Ivermectin against Infectious Bovine Rhinotracheitis Virus in Rabbit Model

**DOI:** 10.3390/ani13203164

**Published:** 2023-10-10

**Authors:** Chen Wang, Yingyu Chen, Xi Chen, Changmin Hu, Jianguo Chen, Aizhen Guo

**Affiliations:** 1The National Key Laboratory of Agricultural Microbiology, Huazhong Agricultural University, Wuhan 430070, China; wangchen2012@webmail.hzau.edu.cn (C.W.);; 2Hubei Hongshan Laboratory, Huazhong Agricultural University, Wuhan 430070, China; 3College of Veterinary Medicine, Huazhong Agricultural University, Wuhan 430070, China; 4Hubei International Scientific and Technological Cooperation Base of Veterinary Epidemiology, Huazhong Agricultural University, Wuhan 430070, China; 5Key Laboratory of Development of Veterinary Diagnostic Products, Ruminant Bio-Products, Ministry of Agriculture and Rural Affairs, Huazhong Agricultural University, Wuhan 430070, China

**Keywords:** ivermectin, infectious bovine rhinotracheitis, bovine herpes virus 1(BoHV-1), treatment, infectious bovine rhinotracheitis virus (IBRV), rabbits

## Abstract

**Simple Summary:**

Infectious bovine rhinotracheitis is responsible for significant economic losses in the cattle industry worldwide, and several countries are working toward controlling or eradicating the infection. This study aimed to investigate whether ivermectin had a therapeutic effect on herpesvirus infections. Firstly, this study showed that low concentrations (6–25 nM) of IVM (ivermectin) can inhibit viral replication in infected cell culture in a dose-dependent way. Importantly, Japanese large-eared white rabbits were infected with 10^7.0^TCID_50_/mL of IBRV viral solution via nasal drops and then treated with different doses of ivermectin at different time points after the attack. Ivermectin treatment could have a therapeutic effect by decreasing clinical signs and viral shedding. IVM could be a candidate drug for early antiviral therapy against IBRV infection.

**Abstract:**

Infectious bovine rhinotracheitis (IBR) caused by bovine herpes virus 1 (BoHV-1) can lead to enormous economic losses in the cattle industry. Vaccine immunization is preferentially used to decrease its transmission speed and resultant clinical signs, rather than to completely stop viral infection. Therefore, a drug effective in treating IBR is urgently needed. Our previous work demonstrated that ivermectin significantly inhibited viral replication in a cell infection model. This study aimed to investigate its antiviral effects in vivo by using a rabbit infection model. The viral inhibition assay was first used to confirm that ivermectin at low concentrations (6–25 nM) could reduce viral titers (TCID_50_) significantly (*p* < 0.001) at 24 h post-infection. In rabbits, ivermectin was administrated with one to three doses, based on the recommended anti-parasite treatment dosage (0.2 mg/kg bodyweight) through subcutaneous injection at different days post-infection in the treated IBRV infection groups, while non-treated infection group was used as the control. The infected rabbits showed hyperthermia and other clinical signs, but the number of high-fever rabbits in the ivermectin treatment groups was significantly lower than that in the non-treated infection group. Furthermore, in ivermectin treatment groups, the cumulative clinical scores correlated negatively with drug doses and positively with delay of administration time post-infection. The overall nasal shedding time in ivermectin-treated groups was two days shorter than the non-treated challenge group. At the same time point, the titer of neutralizing antibodies in the treatment group with triple doses was higher than the other two-dose groups, but the difference between the treatment groups decreased with the delay of drug administration. Correspondingly, the serious extent of lung lesions was negatively related to the dosage, but positively related to the delay of drug administration. The qPCR with tissue homogenates showed that the virus was present in both the lung tissues and trigeminals of the infected rabbits. In conclusion, ivermectin treatment had therapeutic effect by decreasing clinical signs and viral shedding, but could not stop virus proliferation in lung tissues and trigeminals.

## 1. Introduction

Infectious bovine rhinotracheitis virus (IBRV), also called bovine herpesvirus 1 (BoHV-1), is a major viral pathogen of cattle worldwide causing infectious bovine rhinotracheitis (IBR), infectious pustular vulvovaginitis (IPV), and infectious pustular balanoposthitis (IPB) [1] in cattle of all ages and breeds [2,3]. After acute infection, the virus can cause lifelong latent infection [4], and its reactivation, likely induced by certain stressors, leads to virus spread within and among cattle herds. In addition, BoHV-1 can cause host immunosuppression and the destruction of mucosal surfaces in the upper respiratory tract, thereby promoting secondary infection with bacterial pathogens [5] and contributing to bovine respiratory disease complex (BRDC). Vaccines are frequently used to prevent IBR by decreasing its transmission and resultant clinical signs [6]. However, both inactivated and attenuated vaccines have a window for the development of protective immunity. Therefore, there is an urgent need for drugs that can treat IBRV infection in this period.

Ivermectin (IVM) is a 2015 Nobel prize-winning medicine that has been approved as an antiparasitic drug with a broad spectrum. In addition, IVM can inhibit the replication of many DNA viruses, including human and animal herpes viruses such as herpes simplex virus (HSV), pseudorabies virus [7], and BoHV-1 [8]. This laboratory previously demonstrated that IVM at 25 μM could significantly inhibit IBRV replication at cell infection models by stopping viral nuclear import pathways mediated by the import protein, α/β [8]. On the other hand, IVM was shown to inhibit RNA viruses such as severe acute respiratory syndrome-coronavirus 2 (SARS-CoV-2) [9]. An IVM concentration of 5 μM can inhibit this virus in vitro for up to 48 h and could be potentially used to treat human COVID-19 clinically. However, the key issue is that for humans, the maximum total plasma concentration (Cmax) and lung concentration in vivo were 0.0327 μM and 0.0873 μM, respectively, after oral administration of the approved dose of IVM, which were far less than the concentration resulting in a 50% inhibition in vitro (IC50; 2 μM) [10]. Similarly in calves, when the approved anti-parasite dose of IVM (0.2 mg/kg) was subcutaneously administered, the plasma Cmax expressed as area under the concentration–time curve (AUC) was about 0.52 μM (459 ng·d/mL) 4–6 days post-treatment, which was far less than the concentration 25 μM determined by our lab-mates [11].

This report describes an alternative technique to inoculate rabbits and to reproduce infection with bovine herpesvirus 1 and 5 [12]. Vaccine-related experiments have utilized rabbits to assess the immune protective efficacy [13,14,15]. Furthermore, we successfully developed the rabbit infection model, and the results demonstrated that rabbits exhibited respiratory symptoms and that the virus was detectable in their tissue samples. This study aimed to determine whether IVM would display an antiviral effect on BoHV-1 infection in vitro at a low concentration, close to the plasma Cmax post-treatment in vivo by using a rabbit infection model. The findings would be significant to clinically control IBR in cattle by using drugs.

## 2. Materials and Methods

### 2.1. Cells and Culture

The Madin–Darby bovine kidney cell line (MDBK) was purchased from China General Microbiological Culture Collection Center (CGMCC). The cells were cultured in DMEM (Thermo Fisher, Gibco, NY, USA) complete medium with 5% fetal bovine serum (OPCEL, China) and 2% penicillin (final concentration of 200 IU/mL) (Thermo Fisher, USA)—streptomycin (final concentration of 200 µg/mL) solution at 37 °C and 5% CO_2_ atmosphere. For viral growth, the cells were grown in DMEM maintenance medium containing 2% fetal bovine serum and 1% penicillin (final concentration of 100 IU/mL)-streptomycin (final concentration of 200 µg/mL) solution.

### 2.2. Virus and Growth

IBRV HB06 virulent strain was clinically isolated from a diseased calf with IBR by this laboratory, and stored as No. CCTCC V201024 in the Collection Center of China (CCTCC) at Wuhan University, China. For viral replication, the MDBK cells were cultured at more than 90% confluence in DMEM complete medium, inoculated with IBRV HB06 for 2 h, and incubated in DMEM maintenance medium for another 3–5 days until cytopathic effect (CPE) occurred in over 80% of the cell sheets. Then, the cells were frozen three times and clarified via centrifugation at 3000× *g* for 5 min. The supernatant was collected and titrated as TCID_50_/mL according to Reed and Muench [16] described below and stored for cell or rabbit infection.

### 2.3. Effect of IVM on Viral Binding and Entry

MDBK cells were inoculated with 0.1 MOI of IBRV HB06 in presence of 6 nM, 12.5 nM, and 25 nM of IVM (APExBIO, USA) prepared according to the manufacturer’s instructions [8,17], while the control cells were inoculated with the same MOI but without IVM for 2 h at 37 °C. Then, the wells were washed three times with PBS to remove the excess virus and IVM, maintenance media were added and they were further incubated for 16 h. The culture samples were collected and the virus concentration was titrated.

### 2.4. IVM Viral Inhibition Assay

The MDBK cells in a 96-well plate were inoculated with IBRV HB06 of 0.1 MOI for 2 h in an incubator at 37 °C and 5% CO_2_. Then, the plates were washed with PBS (pH 7.0) to remove the unabsorbed virus. After washing, the maintenance medium was added with IVM at three final concentrations (6 nM, 12.5 nM, and 25 nM) and kept in the wells for 16 h, 24 h, and 36 h. The wells with the infected cells but without IVM were set as the negative control. The cell culture samples were, respectively, collected from each well at 16 h, 24 h, and 36 h post-infection (PI) for virus titration.

### 2.5. Effect of IVM on Cytokine Production by Infected Cells

The levels of the cytokines, IFN-γ, IL-4, and IL-8, in the cell culture supernatants collected at different time points were measured by using Bovine IFN-γ ELISA kit (EBC101g, NEOBIOSCIENCE, Shenzhen, China), Bovine IL-4 ELISA kit (EBC006, NEOBIOSCIENCE, Shenzhen, China), and Bovine IL-8 (CXCL8) ELISA kit (EBC08, NEOBIOSCIENCE, Shenzhen, China).

### 2.6. Animal Experiment

This report describes an alternative technique to inoculate rabbits and to reproduce infection with bovine herpesvirus 1 and 5 [12]. We established a rabbit infection model in the early stage. The results showed that clinical symptoms such as high fever, eye discharge, and nasal discharge appear after infection (Appendix A). The nasal shedding can persist for a long time, lesions can be observed in the lung tissue, and pathogens can be detected in the lung and trigeminal ganglion.

The protocol (HZAURAB-2023-0007) regarding to this animal infection was approved by the Committee on the Ethics of Animal Experiments at Huazhong Agricultural University and conducted in strict accordance with the Guide for the Care and Use of Laboratory Animals, Hubei Province, China.

Thirty-three female Japanese white rabbits of about 1.5 kg were purchased from and raised in Laboratory Animal Center at Huazhong Agricultural University. They were randomly divided into 11 groups. Each of Group 1–10 was intranasally (IN) infected with IBRV HB06 (107 TCID_50_/mL), but only Groups 1–9 were administered with IVM via hypodermic injection at three different doses with one dose equal to the treatment dose 0.2 mg/kg recommended by the manufacturer’s instructions, and double (0.4 mg/kg) and triple (0.6 mg/kg) doses on different days post-challenge (DPC) (0, 3, 7 DPC). Group 11 was mock-treated with DMEM as the blank control (Table 1).

### 2.7. Clinical Sign Observation and Scoring

The clinical signs of the test rabbits were observed and recorded, mainly including mental status, nose discharge, eye discharge, congestion or bleeding of the nasal mucosa, conjunctivitis, and cough. They were further scored according to the criteria listed below in Table 2.

### 2.8. Neutralization Antibody Test

The protocol provided by World Organization for Animal Health (WOAH) Manual of Diagnostic Tests and Vaccines for Terrestrial Animals was employed to describe the test procedure for the serum samples. The serum samples obtained at 0, 7, 14, 21, and 28 dpi were tested for SN antibodies. The tested serum was first inactivated at 56 °C for 30 min, and serially diluted 10-fold from 2^−1^–2^−8^. Then, 50 μL of diluted serum was mixed with equal volume of 200 TCID_50_/0.1 mL of IBRV HB06 in one well with each dilution for four wells in 96-well microtiter plates. The mixed samples were incubated at 37 °C for 1 h, 100 μL per well of MDBK cell suspension was added, and CPE was observed for 2–4 days. The neutralizing antibody titer of the tested sera was defined as the maximum serum dilution which was capable of preventing 50% of the infected cell cultures from developing CPE.

### 2.9. Evaluation of Lung Lesions

The rabbits were euthanized 28 d after the viral challenge, and a postmortem was undertaken. Furthermore, lung tissue in 1 cm^3^ blocks for each animal was taken at the boundary site between the lesion and healthy tissues, fixed with 10% neutralized paraformaldehyde solution, and sent to Wuhan Google Bio for tissue sectioning and hematoxylin-eosin (HE) staining. Histopathological changes were observed and recorded as micrographs under the light microscope.

### 2.10. Primer Design

The qPCR method for detection of IBRV gB gene described in WOAH Manual of Diagnostic Tests and Vaccines for Terrestrial Animals was employed, and the genes and probe are shown below (Table 3).

### 2.11. Statistical Analysis

All results were plotted using GraphPad Prism version 8.3.1 for Windows (GraphPad Software, Boston, MA, USA). Virus titers in the different groups were evaluated via the unpaired Student’s *t*-test. Cytokine levels and clinical signs were evaluated using a two-way ANOVA. Temperature changes and nasal virus shedding were evaluated via Chi-square test. One-way ANOVA was also used for analyzing the neutralizing antibody data.

## 3. Results

### 3.1. Effects of IVM’s Inhibition of IBRV Replication in Cultured Cells

#### 3.1.1. IVM Delayed CPE Appearance

The cell survival results show that low concentrations of ivermectin have no significant effect on cell survival (Appendix A). IVM treatment was able to significantly prolong CPE appearance in the cells in a dose-dependent way. Without IVM treatment, the cells showed significant CPE at 24 h PI. As IVM concentrations increased, the time for CPE emergence was increased. When IVM was at 25 nM, only partial cells demonstrated CPE at 48 h PI (Figure 1). On the contrary, the blank control group did not show any CPE.

#### 3.1.2. IVM Did Not Affect Viral Binding and Entry

The viral titers of different groups treated with 6 nM, 12.5 nM, and 25 nM IVM during the adsorption stage were 10^5.603±0.055^TCID_50_/0.1 mL, 10^5.444±0.193^TCID_50_/0.1 mL, 10^5.933±0.231^TCID_50_/0.1 mL, and 10^5.556±0.096^TCID_50_/0.1 mL, respectively. They were close to the viral titer 10^5.556±0.120^TCID_50_/0.1 mL of the non-IVM treated control group (*p* > 0.05), indicating that IVM did not affect IBRV binding and entry into MDBK cells.

#### 3.1.3. IVM Inhibited Viral Replication

The virus inhibition assay showed that IVM’s inhibitory effect on IBRV replication was time- and dose-dependent (Figure 2). At 16 h PI of IBRV at MOI of 0.1, there was no significant difference between the 6 nM IVM-treated group and non-IVM control group (*p* > 0.05), but both the 12.5 nM and 25 nM IVM-treated groups showed a significant difference compared to the non-IVM control (*p* < 0.05). The viral titer of IBRV decreased from 10^5.524±0.041^TCID_50_/0.1 mL to 10^5.067±0.115^TCID_50_/0.1 mL in the 12.5 nM IVM group, to 10^4.700±0.173^TCID_50_/0.1 mL in the 25 nM IVM group (*p* < 0.05) (Figure 2a).

At 24 h PI, all three IVM-treated groups displayed significantly lower viral titers than the non-IVM control group (*p* < 0.05). The inhibitory effect of the 6 nM IVM group climbed approximately to the level of the 12.5 nM IVM group (*p* > 0.05), and there were significant differences among 6 nM, 12.5 nM, and 25 nM-treated groups (*p* < 0.05). The 6 nM and 12.5 nM IVM treatment decreased IBRV titer from 10^7.444±0.096^TCID_50_/0.1 mL in the non-IVM control group to 10^6.647±0.132^TCID_50_/0.1 mL and 10^6.588±0.137^TCID_50_/0.1 mL, respectively, while 25 nM IVM treatment further decreased IBRV titer to 10^5.822±0.168^TCID_50_/0.1 mL (Figure 2b).

After 36 h PI, although there was no significant difference among the 6 nM and 12.5 nM IVM-treated groups and the non-IVM control group (*p* > 0.05), the 25 nM IVM-treated group still showed very significant differences in inhibition of viral replication (10^6.468±0.108^TCID_50_/0.1 mL) compared to the non-IVM control group (10^7.577±0.011^TCID_50_/0.1 mL) (*p* < 0.001) (Figure 2c).

Taken altogether, in the IBRV cell infection model for three timepoints post-infection, 25 nM displayed significant inhibition on viral infection at 16 h, 24 h, and 36 h PI, while IVM 12.5 nM displayed significant inhibition at 16 h and 24 h PI, and IVM 6 nM, only at 24 h PI.

#### 3.1.4. IVM’s Effects on Cytokine Production in IBRV Infected Cells

The culture supernatants were collected at 6 h, 12 h, 18 h, 24 h, 36 h, and 48 h PI, and the levels of IFN-γ, IL-4, and IL-8 expression were measured using commercial ELISA cytokine assay kits. IL-8 levels in the 25 nM IVM-treated groups were significantly higher than other virus-infected groups at 12 h, 18 h, 24 h, 36 h, and 48 h PI (Figure 3c). For IL-4, there was a significant difference only between the 25 nM IVM-treated group at 48 h PI and the non-IVM control group (Figure 3b). However, there was no significant difference in IFN-γ levels among the different IVM dose groups at different time points (Figure 3a). 

### 3.2. IVM’s Effects on IBRV Infection in Rabbits

#### 3.2.1. Temperature Change of the Experimental Rabbits

After the IBRV HB06 challenge, all test groups (1 to 10) showed fevers of different degrees. The temperature was tested for each animal for 14 days after the IBRV challenge, generating a total of 33 temperature values. The numbers of rabbits with fever in each IVM dose group and administered time group were compared and are shown in Table 4. The blank control group (Group 11) did not have any animals with fever for all tests. Regarding IVM’s administration time, there was no difference between 0 and 3 DPC (*p* > 0.05), and 3 and 7 DPC (*p* > 0.05), but a significant difference between 0 and 7 DPC (*p* < 0.05). As for the IVM dosage, although the higher dose caused a lower number of rabbits with fever, there was no difference between the doses (*p* > 0.05). Lastly, the IVM-treated groups (Groups 1–9) and non-IVM Group 10 were compared; there was a significant difference at all three timepoints between the IVM-administered and non-IVM groups, with different *p* values of 0 DPC (*p* < 0.0001), 3 DPC (*p* < 0.001), 7 DPC (*p* < 0.05). In addition, there was also a significant difference between the groups for all three doses and the non-IVM group with different *p* values of triple doses (*p* < 0.001), double doses (*p* < 0.05), and single-dose (*p* < 0.05).

#### 3.2.2. Clinical Signs of Infected Rabbits

The clinical signs of the rabbits with and without IBRV infection and IVM treatment were observed daily, recorded, and scored for 14 days post-challenge. The blank control group, Group 11, did not show any abnormal signs, and all IVM-treated infected groups did not develop cough and die. However, all infected rabbits showed eye and nasal discharge, along with occasional nasal mucosal bleeding to a different degree. In IVM-treated groups, Group 1 showed mild signs, Groups 7, 8, and 9 showed more severe signs of nasal discharge, while the non-IVM-treated infected group displayed the most serious signs. The clinical scores of each test group are summarized and compared in Table 5. The two-way ANOVA analysis indicated that in three IVM dose groups, only the 0.6 mg/kg used at 0 DPC generated a significantly lower score than that of this dose at 3 DPC (*p* < 0.05). Furthermore, there was a significant difference between the 0.6 mg/kg used in DPC 0 (*p* < 0.001), 0.4 mg/kg used at 3 DPC (*p* < 0.001), 0.2 mg/kg used at 3 DPC (*p* < 0.05), and the non-IVM treated but infected group 10. 

#### 3.2.3. Nasal Virus Shedding

Nasal swabs were collected consecutively within 14 d after IBRV challenge for virus detection with qPCR. The results showed that the virus could be detected for all IVM-treated groups (1 to 9) until 9 DPC, two days earlier than 11 DPC for the non-IVM-treated infected group, Group 10. However, there was no difference in proportions of positive samples, and viral titers between IVM-treated and non-IVM-treated but infected groups using the Chi-square test (*p* > 0.05) (Table 6).

#### 3.2.4. Neutralization Antibody Titers

With IVM administered at different doses and at different time points after the IBRV challenge, the triple-dose group showed slightly higher levels of neutralizing antibodies than the other two-dose groups; as the IVM concentrations increased, the neutralizing antibody levels became higher (Appendix A), but there was no significant difference in neutralizing antibody levels between the groups, according to the One-way ANOVA test (*p* > 0.05).

#### 3.2.5. Gross Lesions and Histopathological Change of Lung Tissues

There were no gross lesions in the lungs of the blank control group, Group 11. However, in the non-IVM-treated but infected Group 10, the lungs showed severe, “flesh-like” lesions, with only some white spots (Figure 4). When IVM was administered 0 DPC, the lungs of Group 1, receiving the highest dose, did not show any obvious lesions. But there were some white spots on some of the rabbits’ lungs, which decreased as the dose increased. When IVM was administered 3 DPC, a similar pattern of lung lesions was observed, but the single-dose group showed “flesh-like” lesions in their lungs. When IVM was administered 7 DPC, white spots appeared on the lungs of all the infected rabbits. 

Furthermore, all the experimental rabbits showed different degrees of pathological change, including infiltration of neutrophils, lymphocytes, and erythrocytes, pulmonary interstitial hyperplasia, and thickening of alveolar walls. The infiltration of these inflammatory cells gradually increased as the IVM doses decreased 0 DPC; when IVM was administered 3 DPC, the alveolar lumens became narrower in addition to infiltration of the inflammatory cells; when IVM was administered 7 DPC, these changes became more serious. In contrast, the non-IVM-treated but infected group showed massive infiltration of inflammatory cells and degraded alveolar epithelial cells, marked thickening of the alveolar walls, and narrowing of the alveolar lumens. Meanwhile, there was no significant change in the alveolar structure for the blank control group (Figure 5).

#### 3.2.6. Detection of the Virus in Tissues Detected with qPCR

Lung tissues and trigeminal ganglions were collected from rabbits from different groups, and the virus was detected with qPCR (Table 7). The results showed that IVM treatment did not affect virus presence in the trigeminal ganglions, indicating development of IBRV latent infection.

## 4. Discussion

IVM is called a wonder drug. Besides the broad spectrum of antiparasitic activity, its antiviral effect on at least 4 DNA viruses and 14 RNA viruses in vitro and in vivo has been reported [17]. In in vitro study, IVM was usually used at sub-cytotoxic concentrations to test its antiviral effect. For example, IVM at concentrations of 2.5 μM and 5 μM could reduce the titers of three different FMD virus serotypes by 2 to 3 logs [19]. IVM at 25 μM displayed antiviral activity against HIV and the dengue virus in infected cell systems through the inhibition of nuclear protein import mediated by Impα/β1 [20]. IVM could inhibit PRRSV infection in PAM-pCD163 cells at 5 μM, 10 μM, and 15 μM in a dose-dependent manner [20]. As mentioned before, IVM at 5 μM could in vitro cause a ~5000-fold reduction in viral RNA at 48 h. IVM could inhibit DNA viruses such as human and animal herpes viruses. Regarding the bovine herpes virus, this laboratory previously demonstrated that IVM at 25 μM could inhibit IBRV replication in infected MDBK cells related to Impα/β1 activity. In addition, IVM could inhibit infection of other viruses responsible for bovine respiratory diseases including BCoV, BPIV-3, BVDV, and BRSV at the concentrations of 2.5 and 5 μM [21]. 

However, in humans and animals administered IVM at a single approved dose, the concentrations of plasma and targeted IVM would be much less than the above concentrations displaying antiviral activities in vitro. Usually, the blood levels of IVM at safe therapeutic doses are in the range of 20–80 ng/mL (0.023–0.914 μM). For example, when IVM was orally administered to humans with a single labeled dose of 200 μg/kg, the peak plasma and lung Cmax was predicted to be only 0.0327 μM (0.0228, 0.0429) and 0.0873 μM (0.0609, 0.115), respectively [22]. In calves, IVM was subcutaneously injected at a single dose of 200 μg/kg, and the mean plasma Cmax (n = 4) in the first 15 days post-treatment was 35 ng/mL (0.040 μM). In sheep, after subcutaneous (s.c.) administration at 200 μg/kg, the mean plasma Cmax (n = 6) was 25.76 ng/mL (or 0.029 μM) [23]. In rabbits, when IVM was subcutaneously injected at a single dose of 300 μg/kg, the parent molecule of IVM was detected in plasma between 1 h and 20 days, and the mean Cmax (n = 5) was 32.02 ng/mL (or 0.037 μM) [24]. 

With regard to the main difference between IVM concentrations for antiviral activities in in vitro cell culture systems and plasma levels after the administration of approved doses for antiparasitic treatment, the safety and the preventive or therapeutic efficacy of IVM for clinical use in humans—against COVID-19 or other human and animal viral infections—are debatable. However, some in vivo experiments have shed light on IVM’s clinical applicability as an antiviral agent. Arévalo used a single dose of 500 µg/kg IVM to treat BALB/c mice infected with 6000 PFU of mouse hepatitis virus (MHV-A59), a type 2 family RNA coronavirus similar to SARS-CoV-2, and concluded that IVM diminished the MHV viral load and disease in the mice [25]. IVM with 1 mg/kg orally administered to hamsters prior to SARS-CoV-2 infection was associated with decreased weight loss, pulmonary inflammation, pulmonary viral titers, and mRNA expression levels of pro-inflammatory cytokines associated with severe COVID-19 disease, and it rapidly induced the production of virus-specific neutralizing antibodies in the late stage of viral infection, indicating that IVM may be effective in reducing the development and severity of COVID-19 in a hamster model of SARS-CoV-2 infection [26]. A systematic review and meta-analysis, to assess the currently available data on the therapeutic potential of IVM for COVID-19 treatment as an additional therapy, revealed that adding IVM led to a significant clinical improvement compared to usual therapies (OR = 1.98, 95% CI: 1.11, 3.53, *p =* 0.02) [27].

Based on the above previous studies and arguments on IVM’s antiviral activities in vitro and in vivo, this study firstly confirmed IVM’s antiviral activity on IBRV in an MDBK cell line after decreasing the IVM concentration from the previous 6 μM, 12.5 μM, and 25 μM to 6 nM, 12.5 nM, and 25 nM, which are in the range of plasma concentrations after administration at the approved dose. To further validate its therapeutic potential in clinical trials, the IBRV rabbit infection model was used. We compared effect of IVM treatment from different dimensions including clinical signs, post mortem examination of gross pathology, lung histopathology, viral detection for nasal swabs, lung tissues, and trigeminal ganglions, among three timepoints and three doses (the drug label dose, double, and triple doses) (Groups 1 to 9) by using non-IVM-treated infected rabbits (Group 10) and non-IVM-treated and non-infected rabbits (Group 11) as controls. The results showed that there was a significant difference between 0 and 7 DPC, indicating the significance of early use of IVM. As described previously, at 1 h after IVM administration, the plasma IVM could be detected, meaning that it was possibly exerting its antiviral activity. Furthermore, it was found that the triple doses of IVM (0.6 mg/kg) administered 0 DPC had the lowest clinical sign scores, suggesting that IVM would display its antiviral activity in a dose-dependent way. Accordingly, the nasal viral shedding time detected via PCR was shorter in all IVM-treated groups than the non-IVM-treated but infected group. The gross pathological and histopathological changes supported the above findings that IVM decreased the pathological lesions and tissue inflammation. 

To investigate the effects of IVM treatment on immunity regulation [17], we found that IVM treatment at 25 nM was consistently associated with significantly greater production of IL-8 by the IBRV-infected cells, showing an increase in the innate immunity in infected cells associated with IVM. In the rabbits, although there was no significant difference in neutralization antibody levels, the fewer doses IVM administered at delayed times post-challenge, the more serious the infiltration of inflammatory cells such as neutrophils and lymphocytes. However, the qPCR results showed that IVM treatment did not affect viral presence in trigeminal ganglions, implying that IVM cannot stop viral latency [28,29]. However, before IVM is considered to be a potential candidate drug for antiviral therapy against IBRV infection, the safety and efficacy of IVM in cattle for IBR treatment should be investigated.

## 5. Conclusions

This study has shown that low concentrations (6–25 nM) of IVM can inhibit viral replication in infected cell culture in a dose-dependent way. In rabbits, IVM at one to three approved therapeutic doses can reduce the nasal virus shedding period, clinical signs, gross pathological and histopathological lesions in an administered dose and time-dependent way. Since the strongest effect was observed on the day the rabbits were infected, IVM could be a candidate drug for early antiviral therapy for IBRV infection.

## Figures and Tables

**Figure 1 animals-13-03164-f001:**
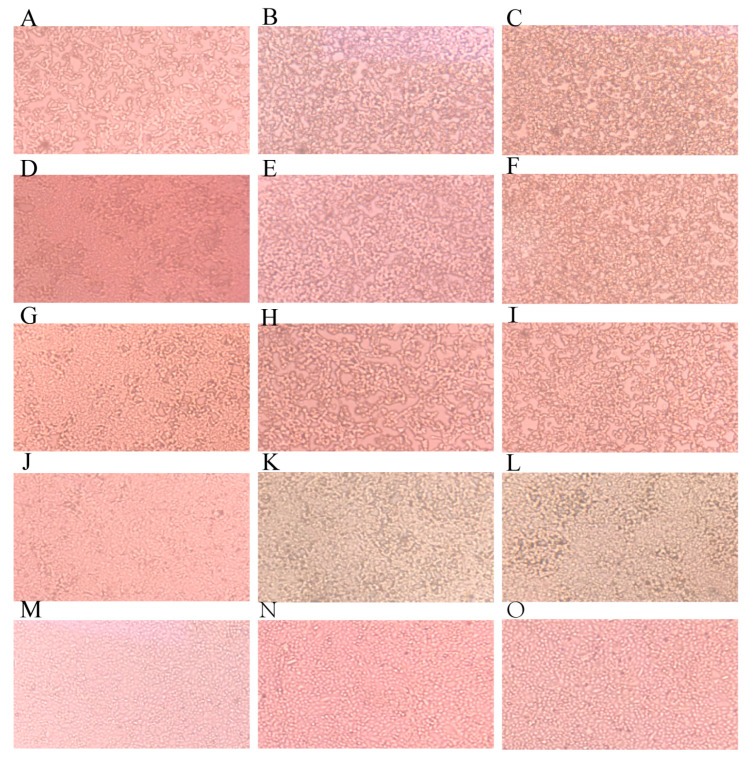
Effects of IVM inhibition on CPE development in IBRV-infected cells (40×). (**A**–**C**) 0.1 MOI IBRV + 0 nM IVM; (**D**–**F**) 0.1 MOI IBRV + 6 nM IVM; (**G**–**I**) 0.1 MOI IBRV + 12.5 nM IVM; (**J**–**L**) 0.1 MOI IBRV + 25 nM IVM; (**M**–**O**) the uninfected and non-IVM-treated control. The first, second, and third columns represent the cells for each treatment at three time points of 24 h, 36 h, and 48 h post-infection, respectively.

**Figure 2 animals-13-03164-f002:**
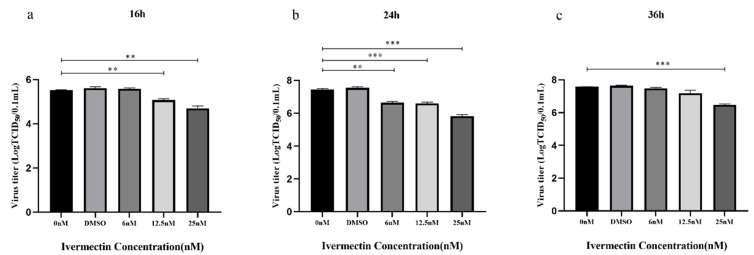
The ivermectin (IVM) antiviral activity at three doses and time points post-infection with IBRV. (**a**–**c**) Represent IBRV titers in the cells treated with IVM at three doses at 16 h (**a**), 24 h (**b**), and 36 h (**c**) post-infection, respectively, after IBRV infection at MOI of 0.1. Means ± SD of n = 3 independent experiments. *** *p* < 0.001, ** *p* < 0.01.

**Figure 3 animals-13-03164-f003:**
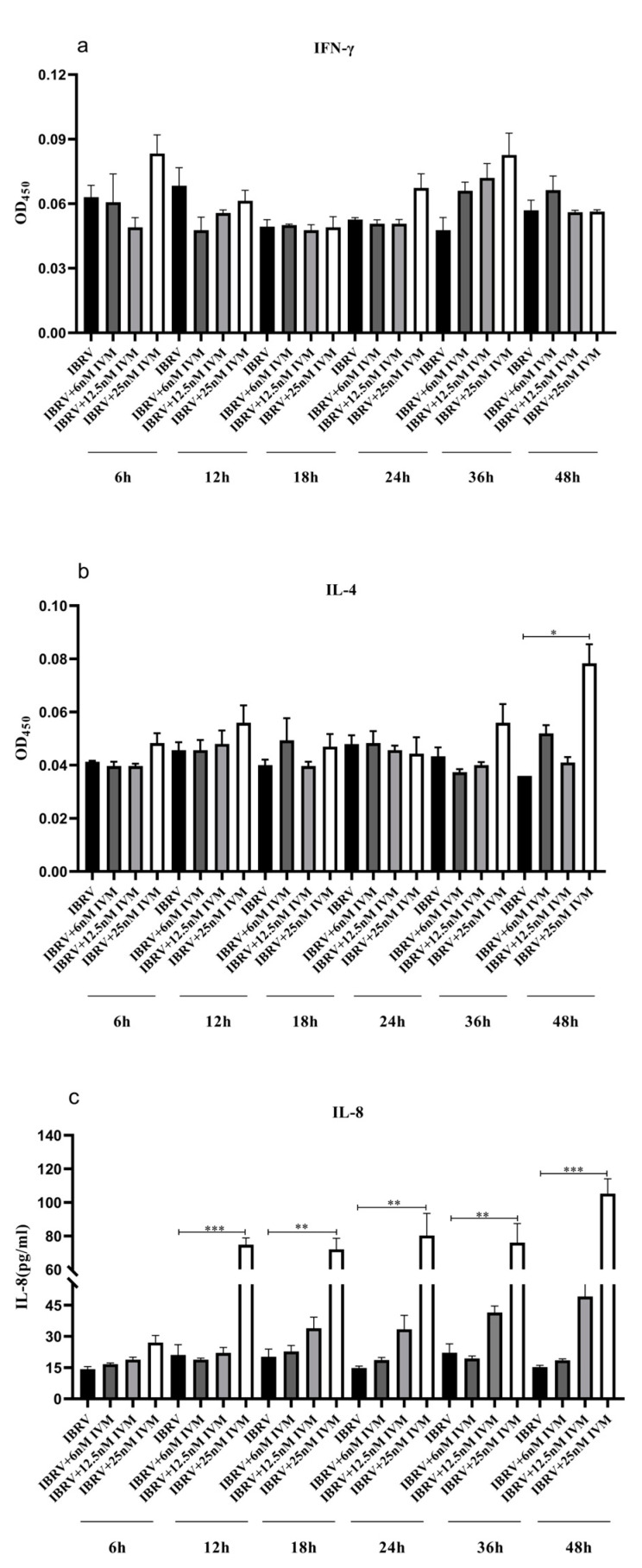
Effects of ivermectin (IVM) on IFN-γ, IL-4, and IL-8 production in IBRV-infected cells. The confluent MDBK cells were infected with IBRV at 0.1 MOI and treated with IVM with different concentrations or mock-treated, and the levels of these three cytokines in cell supernatants were tested at 6, 12, 18, 24, 36, and 48 h PI using commercial cytokine assay kits. Means ± SD of n = 3 independent experiments. *** *p* < 0.001, ** *p* < 0.01, * *p* < 0.05, two-way ANOVA with Tukey’s multiple comparison tests.

**Figure 4 animals-13-03164-f004:**
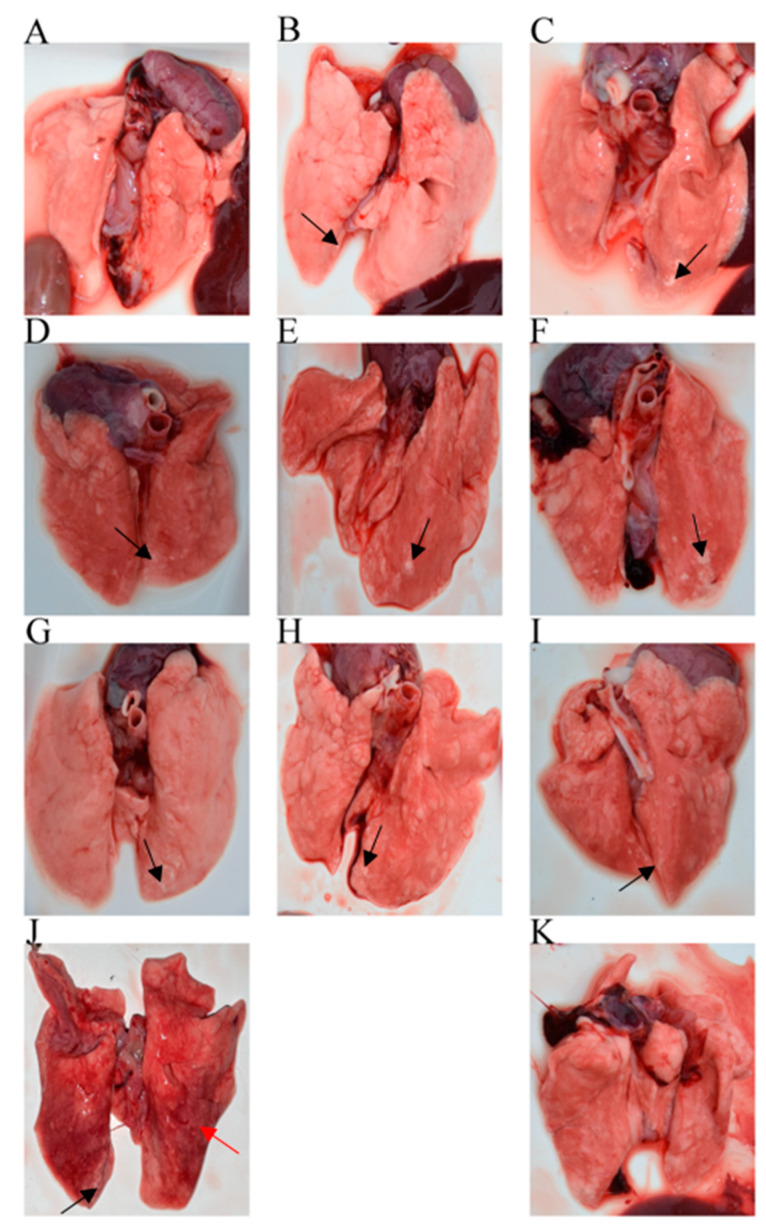
Changes in lung tissues of the experimental rabbits. (**A**–**C**) Different concentrations of ivermectin administered on day 0 (0.6 mg/kg, 0.4 mg/kg and 0.2 mg/kg); (**D**–**F**) Different concentrations of ivermectin administered on day 3 (0.6 mg/kg, 0.4 mg/kg and 0.2 mg/kg); (**G**–**I**) Different concentrations of ivermectin administered on day 7 (0.6 mg/kg, 0.4 mg/kg and 0.2 mg/kg); (**J**) The non-IVM but infected group; (**K**) The blank control group. Black arrow: white spots on this area; Red arrow: “flesh-like” lesions on this area.

**Figure 5 animals-13-03164-f005:**
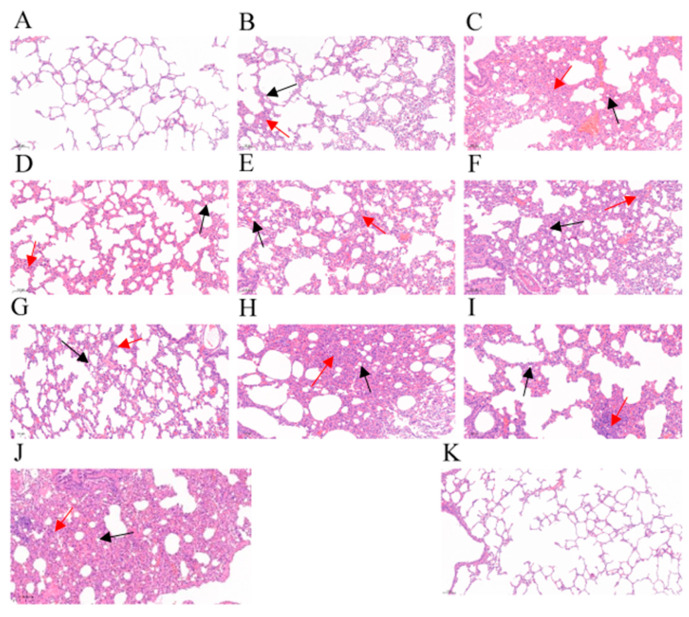
Histopathological changes in lung tissues of the experimental rabbits. (**A**–**C**) Different concentrations of ivermectin administered on day 0 (0.6 mg/kg, 0.4 mg/kg and 0.2 mg/kg). (**D**–**F**) Different concentrations of ivermectin administered on day 3 (0.6 mg/kg, 0.4 mg/kg and 0.2 mg/kg). (**G**–**I**) Different concentrations of ivermectin administered on day 7 (0.6 mg/kg, 0.4 mg/kg and 0.2 mg/kg). (**J**) The non-IVM but infected group. (**K**) The blank control group. Black arrow: narrowing of the alveolar lumens. Red arrow: infiltration of neutrophils, lymphocytes, and erythrocytes.

**Table 1 animals-13-03164-t001:** Experiment design on antiviral effect of ivermectin (IVM) in rabbits.

Groups	N	Inoculums	IVM Doses (mg/kg)	Days of IVM Administered Post Viral Challenge (DPC)
1	3	IBRV HB06, 10^7.0^ TCID_50_/mL, 2 mL	0.6	0
2	3		0.4	0
3	3		0.2	0
4	3		0.6	3
5	3		0.4	3
6	3		0.2	3
7	3		0.6	7
8	3		0.4	7
9	3		0.2	7
10	3		/	/
11	3	DMEM, 2 mL	/	/

**Table 2 animals-13-03164-t002:** Scoring criteria on clinical signs [18].

Observation Item	Scoring Criteria
Nose and eye discharge	None 0, mild discharge 1, moderate discharge with mild mucopurulent discharge 2, moderate mucopurulent discharge 3, severe mucopurulent discharge 4
Mental disorder	None 0, mild 1, moderate 2, severe 3
Nasal mucosa	Normal 0, congestion 1, ulcer 2
Temperature	Increase of 0.5 °C, 1.0 °C, 1.5 °C, and 2.0 °C or more corresponds to 1, 2, 3, 4, respectively

**Table 3 animals-13-03164-t003:** Primers for qPCR detection of IBRV gB gene.

Primers	Primer Sequences (5′-3′)
gB-F	TGTGGACCTAAACCTCACGGT
gB-R	GTAGTCGAGCAGACCCGTGTC
gB-Probe	[FAM]-AGGACCGCGAGTTCTTGCCGC-[TAMRA]

**Table 4 animals-13-03164-t004:** Comparison on the numbers of rabbits with fever infected with IBRV between ivermectin- (IVM) and non-IVM-treated groups.

IVM Doses (mg/kg)	IBRV Challenge	IVM Administration Time (DPC)			Total
		0	3	7	
0.6	+	8/33	13/33	11/33	32/99 **
0.4	+	11/33	8/33	17/33	36/99 *
0.2	+	11/33	10/33	17/33	38/99 *
Total	+	30/99 ***	31/99 **	45/99 *	/
0	+	/			23/33

Note: *, **, *** represent the *p* values of <0.0.5, 0.01, and 0.001 of the statistical difference when the IVM group and non-IVM-treated group were compared via the Chi-square test.

**Table 5 animals-13-03164-t005:** The scores for clinical sign of the rabbits with or without ivermectin (IVM) treatment for IBRV-infected or non-infected animals.

IVM Doses (mg/kg)	IBRV Challenge	Summed Scores at Different IVM Administration Times			Total Scores
		0 DPC	3 DPC	7 DPC	
0.6	+	14 **	43	26	83
0.4	+	26	24 **	39	89
0.2	+	36	24 *	37	97
Total	+	76	91	102	/
0	+	/			62
0	-	/			0

Note: *, ** represent *p* values of <0.05, <0.01, and <0.001 of statistical difference between IVM-treated and non-IVM-treated but infected groups. Compared with two-way ANOVA with Turkey’s multiple comparison tests.

**Table 6 animals-13-03164-t006:** The qPCR assay on nasal viral shedding by the tested rabbits.

IVM Doses (mg/kg)	IBRV Challenge	IVM Administration Time (Positive/Total Samples)			Total Proportions
		0 DPC	3 DPC	7 DPC	
0.6	+	22/42	24/42	23/42	69/126
0.4	+	22/42	23/42	24/42	69/126
0.2	+	23/42	23/42	24/42	70/126
Total	+	67/126	70/126	71/126	/
0	+	/			29/42
0	-	/			0

**Table 7 animals-13-03164-t007:** Proportions of the positive in total samples in IBRV detection with qPCR.

IVM Doses (mg/kg)	IBRV	IVM Administration Time						Total	
		0 DPC		3 DPC		7 DPC			
		L	T	L	T	L	T	L	T
0.6	+	1/3	2/3	1/3	3/3	2/3	1/3	4/9	6/9
0.4	+	1/3	2/3	2/3	1/3	2/3	3/3	5/9	7/9
0.2	+	2/3	2/3	3/3	1/3	3/3	3/3	8/9	6/9
Total	+	4/9	6/9	6/9	5/9	7/9	7/9		
0	+	/						3/3	3/3
0	-	/						0/3	0/3

Note: The L and T represent lung tissues and trigeminal ganglions, respectively. IVM means ivermectin.

## Data Availability

Primary data used in this paper are available from the authors upon request (aizhen@mail.hzau.edu.cn).

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
