# Peer review of "Evaluation of Antiviral Activity of Ivermectin against Infectious Bovine Rhinotracheitis Virus in Rabbit Model"

_animals, 2023, doi:10.3390/ani13203164_

Round 1

Reviewer 1 Report

The aim of this study was to demonstrate, using a rabbit infection model, whether IVM can produce antiviral effects against BoHV-1 infection in vitro at low concentrations close to post-treatment plasma Cmax. The results of this study will have important implications for the clinical use of drugs to control IBR. However, more data are needed to support the viral infection modelling, and the following issues need further clarification.

1. In line 18, the first occurrence of IVM should be in full.

2. The full name of BRD is not bovine respiratory disease syndrome.

3. Clinical symptom scoring criteria for IBRV infection in rabbits were not found in reference [14], please check and add to the literature.

4. What is the basis for IBRV intranasal infection in rabbits? How was the IBRV rabbit infection model demonstrated to be successful? Clinical signs are most evident at how many days after infection? Please specify the method of body temperature measurement. What is the highest body temperature that can be reached? On what days after infection does the body temperature return to normal? The paper lacks a clear description of the post-infection symptoms in rabbits.

5. Please clarify the diluent used for subcutaneous injection of IVM into rabbits.

6. Please add the pathological changes in the nasal mucosa, trachea and bronchi of rabbits after the attack.

7. Figure 1 lacks magnification.

8. Why are the time points for titration of the virus and observation of the cytopathic lesions inconsistent in the Figures 1 and methods 2.4? Please give an explanation.

9. Please clarify the time point for neutralising antibody detection. On what day after viral infection were samples collected for testing?

10. The paper concludes that the best antiviral effect is observed on the day of infection in rabbits. Since the paper provides little information on the rabbit infection model, can it be assumed that the symptoms without medication are also transient? Please provide more informative information on the establishment of the viral infection model.

Author Response

I would like to thank you, the expert reviewers in the field very much for insightful comments and constructive suggestion on our manuscript (MS).

Reviewer 2 Report

This MS in the topic “Evaluation of Antiviral Activity of Ivermectin against Infectious Bovine Rhinotracheitis Virus in Vivo presents” an intriguing topic concerning the use of ivermectin to prevent and inhibit bovine rhinotracheitis virus in vivo, utilizing a rabbit model. Nevertheless, the authors should consider incorporating additional details to enhance the completeness of this manuscript.

1.           The experiment using a rabbit as a model should include an explanation of its relevance in the introduction.

2.           In the abstract, the administration route for Ivermectin should be mentioned.

3.           The author needs to explain all statistical methods used in this investigation in the same location as Statistical analysis, for example the Chi-square test on line 278.

4.           Author should append cytotoxicity test results to this manuscript to explain why each Ivermectin concentration was used in the experiment.

5.           Figures 1 and 3 should be magnified to present clearly.

6.           The Neutralization Antibody titer result must be included in this manuscript, even if it is not statistically significant.

This MS in the topic “Evaluation of Antiviral Activity of Ivermectin against Infectious Bovine Rhinotracheitis Virus in Vivo presents” an intriguing topic concerning the use of ivermectin to prevent and inhibit bovine rhinotracheitis virus in vivo, utilizing a rabbit model. Nevertheless, the authors should consider incorporating additional details to enhance the completeness of this manuscript.

1.           The experiment using a rabbit as a model should include an explanation of its relevance in the introduction.

2.           In the abstract, the administration route for Ivermectin should be mentioned.

3.           The author needs to explain all statistical methods used in this investigation in the same location as Statistical analysis, for example the Chi-square test on line 278.

4.           Author should append cytotoxicity test results to this manuscript to explain why each Ivermectin concentration was used in the experiment.

5.           Figures 1 and 3 should be magnified to present clearly.

6.           The Neutralization Antibody titer result must be included in this manuscript, even if it is not statistically significant.

Author Response

(The authors gave the same response as above.)

Round 2

Reviewer 2 Report

This MS in the topic “Evaluation of Antiviral Activity of Ivermectin against Infectious Bovine Rhinotracheitis Virus in Vivo presents an intriguing topic concerning the use of ivermectin to prevent and inhibit bovine rhinotracheitis virus in vivo, utilizing a rabbit model. This version is accepted for publication.

This MS in the topic “Evaluation of Antiviral Activity of Ivermectin against Infectious Bovine Rhinotracheitis Virus in Vivo presents an intriguing topic concerning the use of ivermectin to prevent and inhibit bovine rhinotracheitis virus in vivo, utilizing a rabbit model. This version is accepted for English.